# Uterine Deletion of *Bmal1* Impairs Placental Vascularization and Induces Intrauterine Fetal Death in Mice

**DOI:** 10.3390/ijms23147637

**Published:** 2022-07-11

**Authors:** Masanori Ono, Natsumi Toyoda, Kyosuke Kagami, Takashi Hosono, Takeo Matsumoto, Shin-ichi Horike, Rena Yamazaki, Mitsuhiro Nakamura, Yasunari Mizumoto, Tomoko Fujiwara, Hitoshi Ando, Hiroshi Fujiwara, Takiko Daikoku

**Affiliations:** 1Department of Obstetrics and Gynecology, Graduate School of Medical Science, Kanazawa University, Takaramachi 13-1, Kanazawa 920-8641, Japan; masanori@tokyo-med.ac.jp (M.O.); n.toyoda@staff.kanazawa-u.ac.jp (N.T.); ktykkn@staff.kanazawa-u.ac.jp (K.K.); thosono@staff.kanazawa-u.ac.jp (T.H.); takeo-m@staff.kanazawa-u.ac.jp (T.M.); rena76jp@med.kanazawa-u.ac.jp (R.Y.); mitsu222@med.kanazawa-u.ac.jp (M.N.); yas1025@med.kanazawa-u.ac.jp (Y.M.); 2Department of Obstetrics and Gynecology, Tokyo Medical University, Shinjuku, Tokyo 160-0023, Japan; 3Division of Animal Disease Model, Research Center for Experimental Modeling of Human Disease, Kanazawa University, Takaramachi 13-1, Kanazawa 920-8641, Japan; 4Department of Cellular and Molecular Function Analysis, Graduate School of Medical Science, Kanazawa University, Kanazawa 920-8640, Japan; h-ando@med.kanazawa-u.ac.jp; 5Division of Integrated Omics Research, Research Center for Experimental Modeling of Human Disease, Kanazawa University, Kanazawa 920-8640, Japan; sihorike@staff.kanazawa-u.ac.jp; 6Department of Obstetrics and Gynecology, Public Central Hospital of Matto Ishikawa, Hakusan 924-8588, Japan; 7Department of Human Life Environments, Kyoto Notre Dame University, Kyoto 606-0847, Japan; fujiwara@notredame.ac.jp

**Keywords:** BMAL1, cKO mouse, miscarriage, NK cell, placental formation, vascularization

## Abstract

Recently, it was demonstrated that the expression of *BMAL1* was decreased in the endometrium of women suffering from recurrent spontaneous abortion. To investigate the pathological roles of uterine clock genes during pregnancy, we produced conditional deletion of uterine *Bmal1* (cKO) mice and found that cKO mice could receive embryo implantation but not sustain pregnancy. Gene ontology analysis of microarray suggested that uterine NK (uNK) cell function was suppressed in cKO mice. Histological examination revealed the poor formation of maternal vascular spaces in the placenta. In contrast to WT mice, uNK cells in the spongiotrophoblast layer, where maternal uNK cells are directly in contact with fetal trophoblast, hardly expressed an immunosuppressive NK marker, CD161, in cKO mice. By progesterone supplementation, pregnancy could be sustained until the end of pregnancy in some cKO mice. Although this treatment did not improve the structural abnormalities of the placenta, it recruited CD161-positive NK cells into the spongiotrophoblast layer in cKO mice. These findings indicate that the uterine clock system may be critical for pregnancy maintenance after embryo implantation.

## 1. Introduction

The uterus is a unique organ specialized for pregnancy, which achieves embryo implantation and fetal delivery. In women, dysmenorrhea is an important clinical sign to show functional or organic disorders of the uterus. Previously, we found that dysmenorrhea is associated with breakfast skipping [1]. Later, a similar positive relationship between breakfast skipping and dysmenorrhea has been shown elsewhere [2,3,4,5]. Recently, we further found that pregnant women who experienced dysmenorrhea at a younger age in the past had a higher incidence of developing hypertensive disorders of pregnancy (HDP) [6]. From these findings, we hypothesized that poor dietary habits during adolescence and young adulthood impair uterine functions and induce latent progression of obstetrics and gynecologic disorders (adolescent dietary habit-induced obstetric and gynecologic disease: ADHOGD) [7].

Since breakfast skipping enhanced starvation stress at the initial stage of the active phase, we proposed that breakfast skipping impairs reproductive functions by disturbing a circadian rhythm via central and/or peripheral clock systems [7,8]. Using a breakfast skipping murine model, we demonstrated that food intake directly regulates the circadian rhythm of the uterine clock [9]. We further found that abnormal starvation stress by irregular uptake of breakfast impaired uterine clock function in mice (in preparation). Consequently, we speculated that the uterine peripheral clock system is influenced by dietary habits and is involved in the mechanism of obstetric and gynecologic diseases.

The circadian oscillation is controlled by clock genes such as *Period* (*Per1*-*3*), *Cryptochrome* (*Cry1*-*2*), *Circadian locomotor output cycles kaput* (*Clock*), and *Brain and muscle aryl hydrocarbon receptor nuclear translocator-like protein 1* (*Bmal1*). Systemic *Bmal1* knockout (KO) female mice were reported to have disorders in multiple organs including multifactorial infertility [10]. However, although circadian expression of clock genes, such as *Per1*-*3*, *Cry1*-*2*, *Bmal1*, and *Clock*, was demonstrated in the murine and rat uterus [11,12], the precise roles of uterine clock genes in reproductive functions remain unclear. Recently, the expression of BMAL1 was demonstrated to be decreased in the endometrium of women suffering from recurrent spontaneous abortion. Researchers also showed that BMAL1-depleted decidual cells inhibited trophoblast invasion, suggesting that the endometrial clock plays an important role in human pregnancy [13].

Based on this background, to investigate the pathological roles of uterine clock genes during pregnancy, we created uterine *Bmal1* cKO mice by crossing *Bmal1*-*loxP* [14] and *progesterone receptor*-*cre* mice [15,16]. In this study, we further examined the effects of uterine *Bmal1*-deletion on uterine functions, especially fertility.

## 2. Results

### 2.1. Uterine Bmal1 cKO Mice Gave No Live Birth

The present study focused on the roles of uterine clock genes in pregnancy events. Thus, we created *Bmal1^f/f^/PR^cre/+^* (cKO) mice crossing with *Bmal1*-*flox* and *PR*-*cre* mice. The loss of BMAL1 in cKO female mice was confirmed by RT-PCR (Figure 1A), Western blotting (Figure 1B), and immunofluorescence (Figure 1C) using non-pregnant cKO and WT (*Bmal1^f/f^/PR^+/+^*) uteruses. To examine the effect of cKO on pregnancy outcome, we mated cKO and WT female mice with fertile C57BL/6 male mice. While seven out of eight WT vaginal plug-positive female mice produced pups, no cKO (n = 15) female mice produced live pups (Figure 1D).

### 2.2. cKO Female Mice Increased Incomplete Miscarriage after Day Six of Pregnancy

To determine the cause of no live birth in cKO female mice, we next examined stage-specific effects of cKO during pregnancy (day one of pregnancy = virginal plug). On day five of pregnancy, the embryo implantation sites in cKO showed no difference in numbers and morphological appearance as compared to WT (Table 1 and Figure 2A). On day six of pregnancy, the number of female mice with implantation and implantation sites was decreased in cKO compared to the WT (57.1% vs. 85.7%, 7.1 ± 0.67 vs. 9.4 ± 0.40, Table 1). The size of the decidual bed seemed smaller in cKO than in WT (Figure 2B).

Among cKO mice on day 8 of pregnancy (n = 30), 11 had no implantation sites, 7 showed resorption sites, and the remaining 12 had normal implantation sites (Table 1 and Figure 2C). There were no differences in the numbers of embryo implantation sites among implantation-positive mice between the cKO and WT. At this stage of pregnancy, branches of the maternal uterine vessels in the mesometrial region become dilated to support fetoplacental development in WT (Figure 2D, green-dotted ellipse); however, vessel dilatation in this region seemed poor in the cKO mice (Figure 2D, red-dotted ellipse). The calculated pixel areas of the blood vessel cavities in the mesometrial region in the cKO seem decreased compared to those in WT, although the difference is not significant (Figure 2E).

### 2.3. cKO Mice Showed Abnormal Placental Formation on Day Twelve of Pregnancy

On day 12 of pregnancy, normal implantation sites were observed in 17 out of 20 (85.0%) WT mice and in 8 (20.5%) out of 39 cKO mice (Table 1 and Figure 3A). Among implantation-positive mice, the numbers of implantation sites in cKO mice were lower than those in WT mice (7.6 ± 1.00 in the cKO group and 9.5 ± 0.46 in the control group, respectively). At this stage of pregnancy, the placenta has three layers: an outer layer of trophoblast giant cells, a middle spongiotrophoblast layer (junctional zone), and the inner labyrinth layer [17]. Maternal vessels contained denucleated red blood cells, whereas fetal vessels still had nucleated erythrocytes. Therefore, we could clearly distinguish maternal vessels from fetal vessels in the placenta using HE staining and found that the placenta of cKO mice had abnormalities in vascular structures. The labyrinth formation was impaired and the areas of maternal vessels became narrowed (Figure 3B). The ratio of calculated areas of maternal vessels (green-dotted areas) among the total areas of both maternal and fetal (red-dotted areas) vessels in cKO mice (Figure 3C) was significantly decreased than those in WT mice (Figure 3D).

### 2.4. cKO Mice Showed Functional Changes in the Uterine Immune Environment

To investigate the key genes related to uterine functional changes in cKO mice, we performed microarray analysis using uterine tissues of WT and cKO mice. In a total of 20,983 genes, up-regulated genes were defined as those with both up-regulated in ZT0 and ZT12 (fold-change > 1; 6147 genes) and were selected by multiplying the absolute value of each WAD (>0.02; 597 genes), whereas down-regulated genes were those with both down-regulated (fold-change < 1; 5169 genes) and were selected (>0.02; 515 genes) as described previously [18]. Gene ontology biological process term enrichment analyses of the identified up- and down-regulated genes detected by WAD were performed using the Database for Annotation, Visualization, and Integrated Discovery (DAVID) with the threshold at a *p*-value < 0.05, and the top 11 groups are presented [18] (Figure 4A). Since the gene ontology terms of immune responses including NK cell activation were dominantly detected in the down-regulated group, we focused on the uterine NK (uNK) cells.

uNK cells are recruited and activated by ovarian hormones, which are dominant immune cells at the maternal-fetal boundary in pregnancy [19]. uNK cells release cytokines or chemokines that induce trophoblast invasion, tissue remodeling, embryonic development, and placentation [20]. We first analyzed the localization of uNK cells using Dolichos biflorus agglutinin (DBA) and periodic Schiff (PAS) staining methods [21,22]. Murine uNK cells are classified into two subsets by DBA and PAS reactivity [23]. DBA-positive (PAS+DBA+) uNK cells produce angiogenic factors [24] and progesterone-induced blocking factor [25], and IFN-γ-producing uNK cells contribute to vessel instability and facilitate pregnancy-induced remodeling of decidual arteries [26,27]. It was reported that DBA-negative (PAS+/DBA-) NK cells exist in the spongiotrophoblast layer, which did not contain DBA-positive uNK cells [25].

On day twelve of pregnancy, abundant DBA-positive uNK cells (Figure 4B, white arrowheads) were detected in the decidua of the WT pregnant mice, whereas DBA-negative (PAS+/DBA-) NK cells (Figure 4B, black arrowheads) were observed in the spongiotrophoblast layer as reported previously [25]. In cKO mice, numbers of DBA-positive and -negative uNK cells were significantly lower than those in WT (Figure 4C). Next, we examined the expression of CD161, a differentiation marker of NK cells, on uNK cells [28]. Recently, CD161 was reported as one of the immune checkpoint molecules on NK cells [29] and its involvement in cancer-associated immunosuppression was proposed [30]. In the WT placenta, CD161 was expressed on uNK cells in both the decidua (upper panel of Figure 4D, white arrowheads) and spongiotrophoblast layers (upper panel of Figure 4D, arrows). In contrast, uNK cells in the decidua layer of cKO mice expressed CD161 (lower panel of Figure 4D, white arrowheads), but its expression was hardly detected in uNK cells in the spongiotrophoblast layer (lower panel of Figure 4D, arrows).

### 2.5. Progesterone Supplementation Rescues Perinatal Outcomes in cKO Female Mice

Since systemic *Bmal1* knockout (KO) mice have infertility accompanied by decreased progesterone (P4) levels due to decreased ovarian function, serum P4 levels in cKO mice were measured. As shown in Figure 5A, on day eight of pregnancy, the P4 level in WT was 65.34 ± 8.04 ng/µL (n = 5). The P4 level in cKO with normal implantation was comparable to WT (48.44 ± 16.96 ng/µL; n = 2); however, that of cKO with resorption was significantly lower (4.75 ± 2.66 ng/µL; n = 4) than WT. On day twelve of pregnancy, the P4 level in WT was 58.06 ± 8.08 ng/μL (n = 4). The P4 level in cKO with live implantation was higher than in WT (88.85 ± 38.29 ng/µL; n = 4); however, the P4 level of the mice without implantation was significantly lower than WT (22.68 ± 20.09 ng/µL; n = 6). Thus, we next examine whether P4 supplementation rescues pregnancy in cKO female mice.

cKO mice with P4 supplementation (n = 5) increased serum P4 levels by more than 145 ng/μL, which was significantly higher than those in WT (Figure 5A). In this condition, the ratio of the mice with normal implantation sites was increased in cKO supplemented with P4 compared to non-treated cKO on both days 8 and 12 of pregnancy (80.0% vs. 40.0% and 57.9% vs. 20.5%, respectively) (Table 1 and Table 2). Notably, in some P4-treated cKO mice (8 out of 12, 66.7%), live pups could be obtained by cesarean section on pregnancy day twenty. In some P4-treated cKO mice, although the implanting embryo showed a normal appearance on day eight of pregnancy, the decidualization of embryo-surrounding decidua was incomplete (Figure 5D).

On day twelve of pregnancy, a severe structural abnormality was observed in the labyrinth layer of P4-treated cKO, which had not been observed in non-treated cKO mice. Although the structure of the labyrinth seems fragile, the areas of maternal and fetal vessels were dilated enough in P4-treated cKO (Figure 5E). The fetuses bearing these malformed placentae would probably have miscarried much earlier if not for P4 supplementation. There were no differences in the ratio of calculated areas of maternal vessels among the total areas of both maternal and fetal vessels between P4-treated cKO and WT mice (Figure 5F).

In the placenta of P4-treated cKO mice, abundant DBA-positive uNK cells (Figure 5G, white arrowheads) were detected in the decidua as observed in those of WT mice. Although DBA-negative (PAS+/DBA-) NK cells (Figure 5G, black arrowheads) were observed in the spongiotrophoblast layer, the numbers were not recovered as compared with non-treated cKO mice (Figure 5H). However, in contrast to non-treated cKO mice, CD161-positive uNK cells were detected in the spongiotrophoblast layer (Figure 5I, arrows).

## 3. Discussion

This study demonstrated that mice lacking BMAL1 expression in the uterus can implant embryos but not sustain a pregnancy. The previous studies showed the possible involvement of *Bmal1* in regulation of male and female reproductive functions [31,32,33]. It was shown that systemic *Bmal1* KO female mice can ovulate but both embryo development and ovarian production of progesterone (P4) are not enough to achieve embryo implantation [10,34]. Later, steroidogenic factor-1 (SF-1) expression-dependent *Bmal1*-deleted female mice (*Bmal1^SF1d/d^*) were demonstrated to fail embryo implantation, which could be rescued by P4 supplementation or normal ovarian transplantation [35], indicating that poor ovarian production of P4 is one of the main causes for infertility of *Bmal1* KO female mice. However, the precise roles of uterine clock genes in pregnancy have been unclear. In this regard, the present study is the first report to show that the uterine clock system is critical for a successful pregnancy.

Histological examinations revealed that cKO mice developed abnormal placental structures. The main feature of placental abnormality in cKO mice was the poor formation of maternal vascular space. Since maternal vascular spaces become narrowed in cKO mice, maternal blood flow into the intervillous spaces in the labyrinth layer is considered to be reduced, suggesting the impaired maternal-fetal nutritional exchanges through this layer. During human placental formation, extravillous trophoblast (EVT) invades the maternal decidua and reconstructs maternal spiral arteries, which is a process essential to maintaining adequate maternal blood flow into the intervillous spaces, by reducing arterial contractility [36]. If this vascular reconstruction is interrupted in the early stage of pregnancy, the subsequent poor blood supply to the intervillous space in the placenta will cause placental dysfunction and induce various pregnancy complications, such as hypertensive disorders of pregnancy (HDP), fetal growth restriction, preterm labor, abortions, and stillbirth in the late stage of pregnancy [37]. In terms of the impaired vascular reconstruction associated with abnormal placental formation, the pathogenesis of our cKO mouse model is corresponding to that of placental dysfunction in human pregnancy complications.

One of the most important properties of the uterus is the acceptance of hemi-allograft during pregnancy. Gene ontology analysis of microarray showed that the immune response genes were down-regulated in the cKO uterus, suggesting the involvement of immune dysfunction in the abnormal placental formation (Figure 3A). It was reported that the cellular immune responses and major cellular subsets, such as regulatory T cells, effector T cells, NK cells, monocytes, macrophages, and neutrophils are involved in the pathogenesis of HDP [38]. Among them, NK cells were shown to directly interact with EVT and regulate their invasion and vascular reconstruction [37]. In accordance with it, GO:0045954 “positive regulation of natural killer cell-mediated cytotoxicity” was detected as down-regulated genes by gene ontology analysis in cKO mice. In the human placenta, it was reported that uNK cells were decreased in placental bed biopsies from women with HDP compared to the control third-trimester decidua, suggesting that altered local cytokine balance may be important in defective trophoblast invasion and spiral artery transformation in these pathological pregnancies [39]. Therefore, we examined uNK distribution and their subtypes around the embryo implantation sites at early pregnancy and in the placenta at mid-pregnancy. In cKO mice, the numbers of NK cells expressing PAS(+)/DAB(-) in the spongiotrophoblast layer decreased and hardly expressed CD161. These findings indicate that the subtype of uNK cells in cKO mice is different from those in WT mice. This also suggests that functional changes in uNK cells were induced by clock gene deletion, especially in the spongiotrophoblast layer where maternal uNK cells directly interact with fetal trophoblast. Considering the immunosuppressive role of CD161 [29], this functional alteration of uNK cells may be critical for trophoblast function to sustain maternal-fetal immune tolerance.

Notably, although this treatment did not completely improve the structural abnormalities of the placenta, pregnancy could be sustained until the end of pregnancy in some cKO mice. It was reported that, although murine uNK cells do not express the classical progesterone receptor, progesterone affects the recruitment and function of uNK cells [25]. In accordance with it, this study showed that P4 supplementation could recruit CD161-positive NK cells into the spongiotrophoblast layer even in cKO mice. It was demonstrated that uNK cells express glucocorticoid receptors and progesterone can regulate NK cells via glucocorticoid receptor [40]. Allowing for a higher concentration of serum P4 in P4-supplemented cKO mice compared to WT mice, the possibility that expression of CD161 on uNK cells in the spongiotrophoblast layer was mediated through a glucocorticoid receptor pathway. As clinical evidence, progesterone supplement was demonstrated to prevent miscarriage in women with recurrent miscarriage of unclear etiology [41] and protect the onset of HDP [42]. Recently, it was also reported that progesterone supplement therapy is clinically effective for the functional failure of the placenta with structural abnormalities [43]. Our cKO model provides experimental evidence in support of this treatment for pregnancy complications.

This study has several limitations. First, we did not show the direct relationship between meal skipping and uterine clock dysfunction-induced miscarriage. Second, since clock genes may have non-clock functions, the deletion of uterine *Bmal1* does not necessarily equal the dysfunction of the uterine clock system. In the future, it is necessary to validate this issue by experimentation using other clock-gene-deleted mice or by constitutively expressed BMAL1 in the uterus.

In conclusion, uterine deletion of *Bmal1* impairs placental vascularization and induces intrauterine fetal death in mice. The maternal vascular spaces in the placenta were reduced in cKO mice, suggesting the presence of placental dysfunction. Since uNK cells in the spongiotrophoblast layer of the placenta did not express CD161, dysfunction in uNK cells is one of the possible mechanisms to induce failure of placental formation in cKO mice. Notably, P4 supplementation recruited CD161-positive NK cells into the spongiotrophoblast layer and rescued pregnant outcomes in some cKO mice. These findings indicate that the uterine clock system is critical for a successful pregnancy and suggest that P4 supplementation can partially recover clock-induced uterine dysfunction. This study also provides experimental evidence that the disorder of a uterine clock system can be a candidate to induce uterine dysfunction during pregnancy in ADHOGD.

## 4. Materials and Methods

### 4.1. Mice

We produced *Bmal1^f/f^/PR^cre/+^* (cKO) mice by crossing *Bmal-loxP* mice (strain #007668, The Jackson Laboratory, Sacramento, CA, USA) [14] and *PR-cre* mice [44]. *PR-cre* mice were kindly provided by Dr. Francesco Demayo (the National Institutes of Health, Bethesda, MD, USA) and Dr. John P. Lydon (Baylor College of Medicine, Houston, TX, USA). C57BL/6-J male mice were purchased from Charles River (Yokohama, Japan) and used for experiments after checking fertility. All experimental procedures and housing conditions were approved by the Kanazawa University Animal Experiment Committee (AP-163729 and AP-214218). All animals were housed and cared for in accordance with the Institutional Guidelines for Experiments using Animals.

### 4.2. Analysis of Pregnancy Events

Female cKO and WT (*Bmal1^f/f^/PR^+/+^*) mice were mated with fertile C57BL/6-J male mice to induce pregnancy (Day 1 = virginal plug). To visualize the implantation sites on days 5 and 6 of pregnancy, 0.1 mL of 1% (*w*/*v*) Chicago Sky Blue 6B (Sigma, St. Louis, MO, USA) in saline was intravenously injected into mice [45]. To examine whether P4 supplementation maintains pregnancy, a Silastic implant (4 cm length × 0.31 cm diameter, Kaneka medical products, Osaka, Japan) containing P4 (Sigma-Aldrich, Saint Louis, MO, USA) was placed under dorsal skin on day 2 of pregnancy [45].

### 4.3. RT-PCR

RNA was extracted from the uteruses of WT and cKO mice with TRIzol Reagent (Thermo Fisher, Waltham, MA, USA) according to the manufacturer’s instructions. After removing contaminating genomic DNA with TURBO DNA-free Kit (Thermo Fisher, Waltham, MA, USA), one µg RNA was reverse-transcribed with Superscript II and oligo(dT) primer to prepare cDNA (Thermo Fisher, Waltham, MA, USA). PCR was performed as previously described [18,46]. The primer sequences for PCR are forward primer (5′-aaagaggcgtcgggacaaaa-3′) and reverse primer (5′-ccatctgctgccctgagaat-3′) for *Bmal1*, and forward primer (5′-ctctcgctttctggagggtg-3′) and reverse primer (5′-tcagtctccacagacaatgcc-3′) for *Rplp0* as an internal control.

### 4.4. Western Blot Analysis

Tissue samples were prepared as previously described [47]. After measuring the protein concentration, extracts with SDS-PAGE sample buffer were boiled for 5 min. Samples were run on 10% SDS-PAGE gel and transferred to a polyvinylidene fluoride membrane. Membranes were blocked with 5% skim milk in Tris-buffered saline Tween 20 and probed with antibodies to BMAL1 (1:2000; Novus Biologicals, Centennial, CO, USA) or ACTIN (1:5000; Santa Cruz Biotechnology, Santa Cruz, CA, USA). Blots were incubated with donkey anti-goat or donkey anti-rabbit IgG conjugated to peroxidase (Jackson ImmunoResearch Laboratories, West Grove, PA, USA). All signals were detected using chemiluminescent reagents (GE Healthcare, Chicago, IL, USA). ACTIN was used as a loading control.

### 4.5. Immunofluorescence

Uteruses harvested from WT and cKO non-pregnant mice were fixed with 4% paraformaldehyde and embedded in OCT after being dehydrated with 30% sucrose. Frozen sections (10 µm) were stained with BMAL1 antibody (1:1000; Novus Biologicals, Centennial, CO, USA) and Hoechst.

### 4.6. Histology and Measurement of Blood Vessel Area

Formalin-fixed paraffin-embedded tissue sections (5 µm) were stained with hematoxylin and eosin for histological analysis. Blood vessel area in the decidua and the placenta were measured by Image—J software on days 8 and 12 of pregnancy, respectively [48].

### 4.7. Immunohistochemistry

Formalin-fixed paraffin-embedded tissue sections (5 µm) were subjected to immunohistochemistry with anti-CD161 rabbit polyclonal antibody (1:2000; Abcam ab197979, San Francisco, CA, USA) after antigen retrieval. The signal was visualized with DAB (Fujifilm Wako Pure Chemical Corporation, Osaka, Japan). Hematoxylin staining was used as counterstaining.

### 4.8. DBA Lectin and PAS Dual Staining

Formalin-fixed paraffin-embedded tissue sections (5 µm) were first stained with DBA lectin and then subjected to PAS staining. In brief, after antigen retrieval, sections were incubated with biotinylated DBA lectin (1:5000; Sigma) and visualized positive cells with DAB. Next, sections were incubated with 1% periodic acid (Fujifilm Wako Pure Chemical Corporation) for 20 min, with Schiff reagent (Fujifilm Wako Pure Chemical Corporation) for 30 min, and 0.5% sodium bisulfite solution (Fujifilm Wako Pure Chemical Corporation, Osaka, Japan) for 5 min. Hematoxylin was used for counter-staining.

### 4.9. Microarray Analysis

Uteruses were collected from matured WT and cKO mice at 9 am and 9 pm. RNA was extracted with an RNeasy mini kit (Qiagen, Hilden, Germany) according to the manufacturer’s instructions. Cyanine-3 (Cy3)-labeled cRNA was prepared from 0.2 µg RNA using a Low Input Quick Amp Labeling Kit (Agilent) according to the manufacturer’s instructions. A total of 0.6 µg of Cy3-labelled cRNA was hybridized to SurePrint G3 Mouse GE microarray 8 × 60 K Ver. 2.0 (G4858A #74809, Agilent Technologies, Santa Clara, CA, USA) for 17 h at 65 °C in a rotating Agilent hybridization oven. After hybridization, microarrays were washed and slides were scanned immediately after washing on an Agilent DNA Microarray Scanner (G2539A) using the one color scan setting for 1 × 60 K array slides. The scanned images were analyzed with Feature Extraction Software 11.0.1.1 (Agilent) using default parameters to obtain background-subtracted and spatially detrended processed signal intensities. Data were normalized and filtered with three filters with GeneSpring software 12 (Agilent Technologies). Differentially expressed genes were extracted by the weighted average difference (WAD) ranking method [49].

### 4.10. P4 Assay

The concentration of P4 in mouse serum was measured by an enzyme-linked immunosorbent assay kit (Cayman, Ann Arbor, MI, USA). All samples were measured in triplicate.

## 5. Statistical Methods

Statistical analyses were performed by unpaired *t*-test (two-sided test). *p* values < 0.05 were accepted as statistically significant.

## Figures and Tables

**Figure 1 ijms-23-07637-f001:**
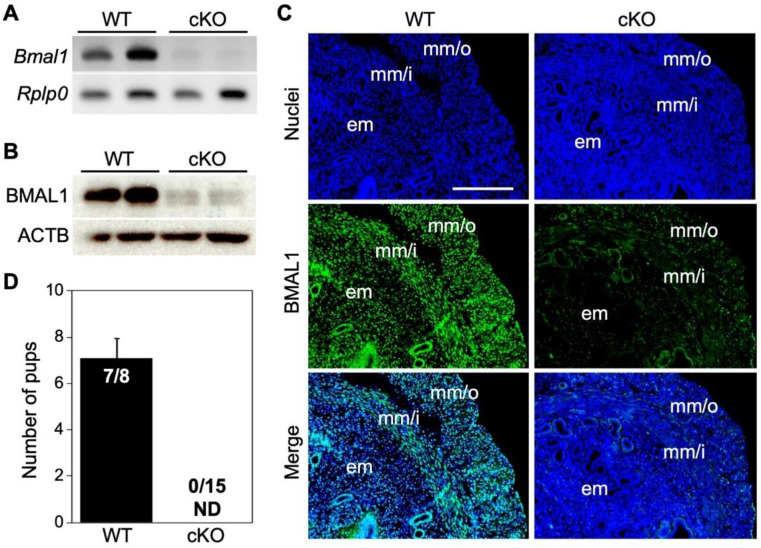
*Bmal1* cKO uteri showed efficient deletion of BMAL1 and no live birth. (**A**–**C**) The uterine deletion of BMAL1 was observed by (**A**) RT-PCR, (**B**) Western blotting, and (**C**) immunofluorescence. (**D**) Uterine *Bmal1* cKO mice showed no live births. The bar shows SE. ACTB, ACTIN; em, endometrium; mm/i, inner circular layer of myometrium; mm/o, the outer longitudinal layer of myometrium; ND, not determined. The scale bar shows 200 µm.

**Figure 2 ijms-23-07637-f002:**
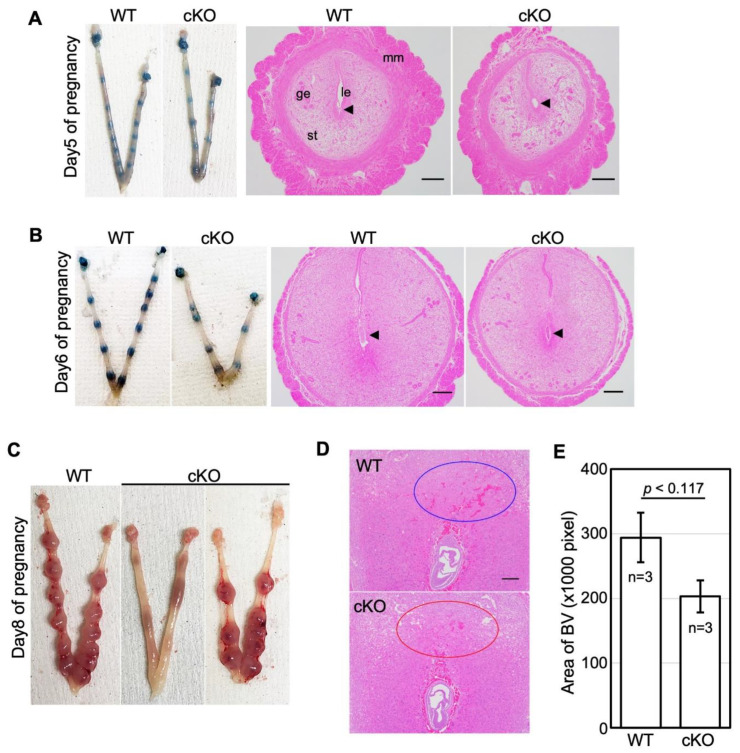
The miscarriage rate is increased in cKO mice after implantation. (**A**,**B**) Representative of uteri and HE-stained section of implantation sites from WT and cKO on days 5 (**A**) and 6 (**B**) of pregnancy. (**C**) Representative of uteri of WT and cKO on day 8. (**D**) HE-stained section of implantation sites and area of the blood vessel (BV) on day 8. (**E**) Although the difference is not significant, the calculated pixel areas of the blood vessel cavities in the mesometrial region in the cKO seem decreased compared to those in WT. The scale bar shows 100 µm. Arrowheads show embryo. Circles show uterine vessels in the mesometrial region. le, luminal epithelium; ge, glandular epithelium; st, endometrial stroma; mm, myometrium.

**Figure 3 ijms-23-07637-f003:**
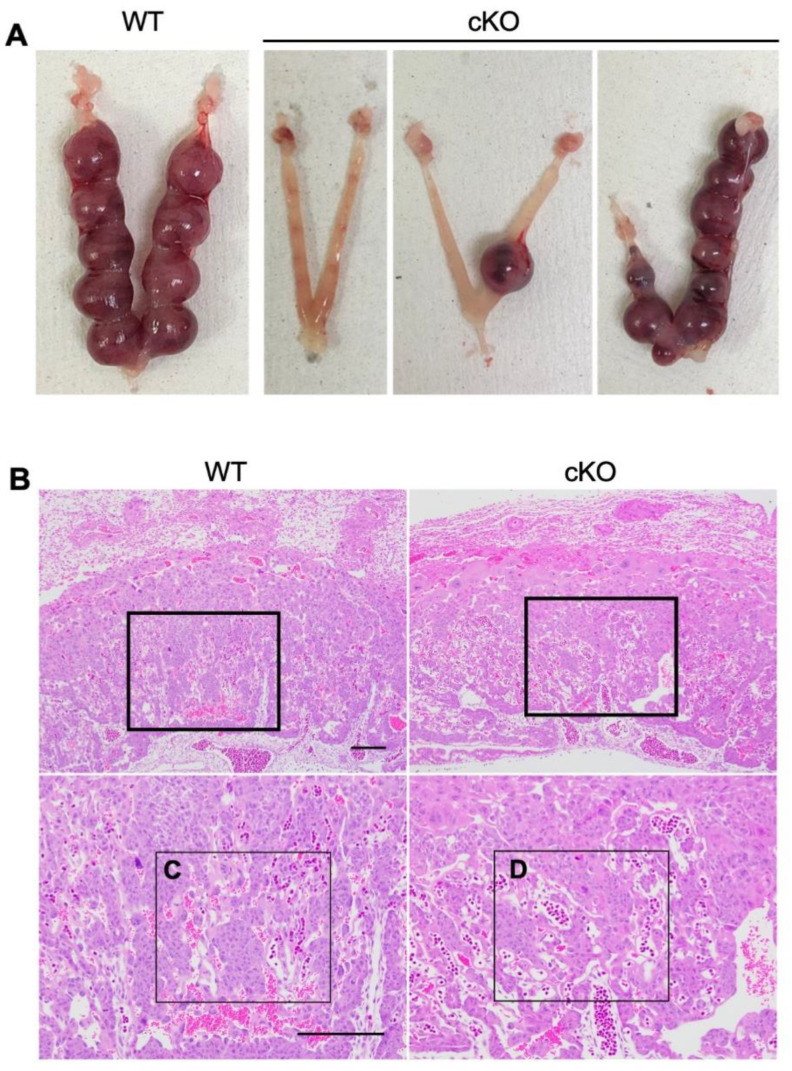
cKO mice show abnormal placental formation on day 12 of pregnancy. (**A**) Representative of uteri from WT and cKO on day 12 of pregnancy. (**B**–**D**) HE-stained section of the placenta. Blue-dotted areas show maternal blood vessels. Red-dotted areas show fetal blood vessels which contain nucleated erythrocytes. (**E**) The percentage of the maternal blood vessels that contains denucleated red blood cells (RBC). The bar shows SE. Differences in the ratio of calculated areas of maternal vessels between the wild-type and cKO mice were analyzed by the unpaired *t*-test. *p* < 0.05. The scale bar shows 100 µm.

**Figure 4 ijms-23-07637-f004:**
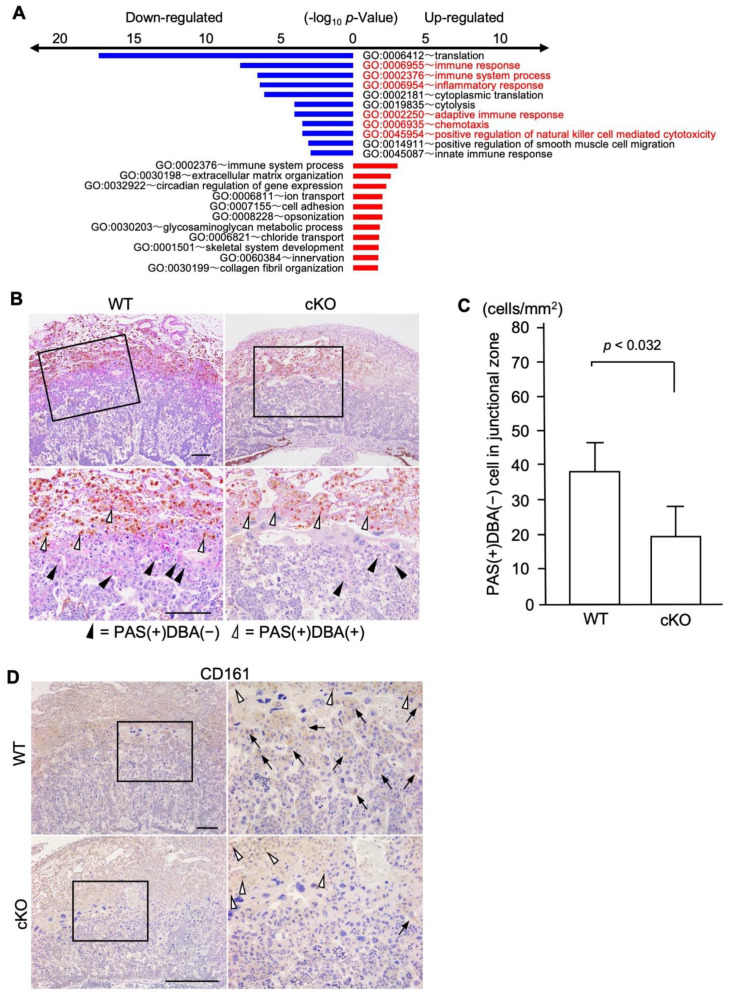
The immune environment in cKO uteri. (**A**) Gene ontology analysis with microarray data. The top 11 groups are shown. (**B**) DBA/PAS double staining with the placenta on day 12 of pregnancy. Black arrowheads show PAS-positive and DBA-negative cells. White arrowheads show both PAS- and DBA-positive cells. (**C**) In cKO mice, numbers of DBA-positive and -negative uNK cells were significantly lower than those in WT. (**D**) CD161 staining with the placenta on day 12 of pregnancy. In the WT placenta, CD161 was expressed on uNK cells in both the decidua (upper panel, white arrowheads) and spongiotrophoblast layers (upper panel, arrows). In contrast, uNK cells in the decidua layer of cKO mice expressed CD161 (lower panel, white arrowheads) but its expression was hardly detected in uNK cells in the spongiotrophoblast layer (lower panel, arrows). Arrows show CD161 positive cells. The scale bar shows 100 µm.

**Figure 5 ijms-23-07637-f005:**
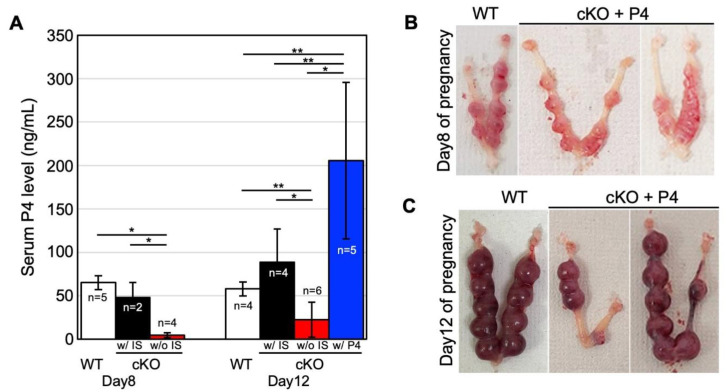
P4 supplement rescues pregnancy failure in cKO mice. (**A**) Serum P4 levels in WT and cKO on days 8 and 12 of pregnancy. The mean P4 levels in the control group (WT, n = 5) and in the cKO groups with normal implantation (w/IS, n = 2) and with confirmed inhalation (w/o IS, n = 4) on day 8 of pregnancy, and those in the control group (WT, n = 4), the cKO groups with implantation (w/IS, n = 4) and without implantation (w/o IS, n = 6), and the progesterone-treated cKO group (w/P4, n = 5) on day 12 of pregnancy. The bar shows STDEV. Differences between groups were analyzed by the unpaired *t*-test. *, *p* < 0.01; **, *p* < 0.05. (**B**,**C**) Representative of the uterus of WT and cKO with P4 supplementation on days 8 (**B**) and 12 (**C**) of pregnancy. (**D**,**E**) Representative of HE-stained uterus of WT and cKO with P4 supplementation on days 8 (**D**) and 12 (**E**). (**F**) The percentage of maternal blood vessels, which contain denucleated red blood cells (RBC). The bar shows SE. Differences in the ratio of calculated areas of maternal vessels between the WT and cKO mice were analyzed by the unpaired *t*-test. (**G**) DBA/PAS double staining of the placenta from cKO with P4 supplementation on day 12 of pregnancy. Black arrowheads show PAS-positive and DBA-negative cells. White arrowheads show both PAS- and DBA-positive cells. (**H**) Numbers of DBA-positive and -negative uNK cells were not significantly different between cKO and cKO witn P4 supplementation. (**I**) CD161 staining of the placenta from cKO with P4 supplementation on day 12 of pregnancy. Arrows show CD161 positive cells in the spongiotrophoblast layers. The scale bar shows 100 µm. N.S., not significant.

**Table 1 ijms-23-07637-t001:** cKO mice start to fail pregnancy maintenance after day six of pregnancy.

Day of Pregnancy	Genotype	No. of Mice	No. of Mice with Normal IS (%)	Mean No. of IS Sites (No. ± SE) *
Day 5	WT	8	100.0	8.3 ± 0.65
	cKO	11	100.0	8.3 ± 0.38
Day 6	WT	14	85.7	9.4 ± 0.40
	cKO	21	57.1	7.1 ± 0.67
Day 8	WT	16	81.3	8.8 ± 0.53
	cKO	30 **	40.0	7.3 ± 0.56
Day 12	WT	20	85.0	9.5 ± 0.46
	cKO	39	20.5	7.6 ± 1.00

* Mean IS numbers among only female mice with normal IS; ** 7 out of 30 mice had resorption. WT; *Bmal1^f/f^/PR^+/+^*, cKO; *Bmal1^f/f^/PR^cre/+^*, IS; implantation site.

**Table 2 ijms-23-07637-t002:** P4 supplementation partially rescues pregnancy of cKO mice.

Day of Pregnancy	Genotype	No. of Mice	No. of Mice with IS (%)	Mean No. of IS Sites (No. ± SE)
Day 8	WT + P4	4	100.0	8.5 ± 0.50
	cKO + P4	10	80.0	8.6 ± 0.46
Day 12	WT + P4	8	100.0	6.5 ± 1.10
	cKO + P4	19	57.9	5.8 ± 0.91

WT; *Bmal1^f/f^/PR^+/+^*, cKO; *Bmal1^f/f^/PR^cre/+^*.

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
