# Peer review of "Uterine Deletion of Bmal1 Impairs Placental Vascularization and Induces Intrauterine Fetal Death in Mice"

_ijms, 2022, doi:10.3390/ijms23147637_

Round 1

Reviewer 1 Report

The paper entitled: Uterine deletion of Bmal1 impairs placental vascularization and induces intrauterine fetal death in mice demonstrates that mice lacking BMAL1 expression in the uterus can implant embryos, but not sustain a pregnancy. This study is original and the topic is interesting enough to attract the readers’ attention. This work could be published after major revision.

.  

My observations are as follows:

·         English revision of the entire manuscript is necessary

·         Cite the number of the ethics committee that gave the okay for this trial

·         Restrict in the abstract to saying what was done and what results were obtained. Considerations should not be put in the abstract

·         In the introduction, in addition to poor eating habits during adolescence and youth that can impair uterine function and induce latent progression of obstetric and gynecological disorders, I would also cite the impact that living in polluted areas can produce. In this regard I would cite the following recent work showing that kallikrein is a marker of environmental exposure in young women:

Could Kallikrein-Related Serine Peptidase 3 Be an Early Biomarker of Environmental Exposure in Young Women? Int J Environ Res Public Health. 2021 Aug 21;18(16):8833. doi: 10.3390/ijerph18168833. PMID: 34444582; PMCID: PMC8392638.

Better explain the abnormalities in vascular structures. “Both vasculatures were irregular and the areas of maternal vessels became narrowed (Fig. 3B). The ratio of calculated areas of maternal vessels (green dotted areas) among the total areas of both maternal and fetal (red dotted areas) vessels in cKO mice (Fig. 3C) was significantly decreased than those in WT mice (Fig. 3D)”.

Explain the consequences of  this finding: “In a total of 20983 genes, up-regulated genes were defined as those with both up-regulated in ZT0 and ZT12 (fold-change >1; 6147 genes) and selected by multiplying the absolute value of each WAD (>0.02; 597 genes), whereas down-regulated genes were those with both down-reg-ulated (fold-change <1; 5169 genes) and selected (>0.02; 515 genes) as described previously [18]”.

Author Response

Thank you for your constructive comments.

In response to your valuable advice, we extensively revised our manuscript as follows.

  1. English revision of the entire manuscript is necessary

We carefully checked the English grammar in the revised manuscript.

  1. Cite the number of the ethics committee that gave the okay for this trial

We added the approved number (AP-163729 and AP-214218) given by the ethics committee in the revised manuscript.

  1.  Restrict the abstract to saying what was done and what results were obtained. Considerations should not be put in the abstract

We deleted the content of considerations from the abstract in the revised manuscript.

  1.  In the introduction, in addition to poor eating habits during adolescence and youth that can impair uterine function and induce latent progression of obstetric and gynecological disorders, I would also cite the impact that living in polluted areas can produce. In this regard I would cite the following recent work showing that kallikrein is a marker of environmental exposure in young women:

Could Kallikrein-Related Serine Peptidase 3 Be an Early Biomarker of Environmental Exposure in Young Women? Int J Environ Res Public Health. 2021 Aug 21;18(16):8833. doi: 10.3390/ijerph18168833. PMID: 34444582; PMCID: PMC8392638.

The recommended paper is very interesting since we also found that other serine peptidase’s expression was upregulated in uterus from both breakfast skipping and the clock gene-impaired mouse models. Therefore, we would like to cite this paper in our next paper that is currently under preparation.       

  1. Better explain the abnormalities in vascular structures. “Both vasculatures were irregular and the areas of maternal vessels became narrowed (Fig. 3B). The ratio of calculated areas of maternal vessels (green dotted areas) among the total areas of both maternal and fetal (red dotted areas) vessels in cKO mice (Fig. 3C) was significantly decreased than those in WT mice (Fig. 3D)”.

We changed the sentence “Both vasculatures were irregular and the areas of maternal vessels became narrowed” to “The labyrinth formation was impaired and the areas of maternal vessels became narrowed” in the revised manuscript.  

  1. Explain the consequences of this finding: “In a total of 20983 genes, up-regulated genes were defined as those with both up-regulated in ZT0 and ZT12 (fold-change >1; 6147 genes) and selected by multiplying the absolute value of each WAD (>0.02; 597 genes), whereas down-regulated genes were those with both down-regulated (fold-change <1; 5169 genes) and selected (>0.02; 515 genes) as described previously [18]”.

We rewrote the next sentence “Gene ontology biological process term enrichment analyses of the identified up-and down-regulated genes detected by WAD were performed…..”.

Reviewer 2 Report

Dear Editor, 

I have provided my comments on this manuscript as below. Currently What I mostly care about is the novelty of this work since a previous report has done a lot of study which covers most of the studies in this work [Proc Natl Acad Sci U S A . 2014;111(39):14295-300], in my point only after this issue the manuscript is addressed by the authors or more data regarding molecular mechanism provided, the manuscript could be considered for further processings and revisions.

Regards,

Jinhu Guo 

-----------------------------------------------------------------------

Title: "Bmal1" should be in italic.

Abstract: 

1) In the beginning, it is better to add one or two brief sentences providing the background.

2) Reproductive defects due to deletion of Bmal1 does not necessarily equal to the function of clock. That is, clock genes may have non-clock functions. This issue need to be discussed in the manuscript. Of course, it is better to have experimental evidence to clarify.

Results

3) The occurrance of Fig.1 panels are not in order, Fig. 1D appears but Fig.1C is absent.

4) "cKOfemalesincreasedincomplete miscarriageafter day 6 of pregnancy", "pregnancy day 12", please phrase this sentence.

Novelty

5) I read the related paper (Proc Natl Acad Sci U S A. 2014 Sep 30;111(39):14295-300) and found that most of the findings in the present manuscript has been covered, including the phynoype description, the histological analysis and supplementation of progesterone. Therefore, I strongly suggest that the authors provide more data on the molecular mechanisms.And this issue is also required to be clearly addressed in the discussion section.

6) "In  accordance  with it,  this  study showed  that P4   supplementation", please remove the redundant space.

7) Some important references are missing, which include:

Endocrinology 2016 Dec;157(12):4914-4929; Front Endocrinol. 2022 Mar 2;13:818272; J Biol Rhythms. 2008 Feb;23(1):26-36.

Author Response

Response to Referee I

Thank you for your constructive comments.

In response to your valuable advice, we extensively revised our manuscript as follows.

  1. English revision of the entire manuscript is necessary

We carefully checked the English grammar in the revised manuscript.

  1. Cite the number of the ethics committee that gave the okay for this trial

We added the approved number (AP-163729 and AP-214218) given by the ethics committee in the revised manuscript.

  1.  Restrict the abstract to saying what was done and what results were obtained. Considerations should not be put in the abstract

We deleted the content of considerations from the abstract in the revised manuscript.

  1.  In the introduction, in addition to poor eating habits during adolescence and youth that can impair uterine function and induce latent progression of obstetric and gynecological disorders, I would also cite the impact that living in polluted areas can produce. In this regard I would cite the following recent work showing that kallikrein is a marker of environmental exposure in young women:

Could Kallikrein-Related Serine Peptidase 3 Be an Early Biomarker of Environmental Exposure in Young Women? Int J Environ Res Public Health. 2021 Aug 21;18(16):8833. doi: 10.3390/ijerph18168833. PMID: 34444582; PMCID: PMC8392638.

The recommended paper is very interesting since we also found that other serine peptidase’s expression was upregulated in uterus from both breakfast skipping and the clock gene-impaired mouse models. Therefore, we would like to cite this paper in our next paper that is currently under preparation.       

  1. Better explain the abnormalities in vascular structures. “Both vasculatures were irregular and the areas of maternal vessels became narrowed (Fig. 3B). The ratio of calculated areas of maternal vessels (green dotted areas) among the total areas of both maternal and fetal (red dotted areas) vessels in cKO mice (Fig. 3C) was significantly decreased than those in WT mice (Fig. 3D)”.

We changed the sentence “Both vasculatures were irregular and the areas of maternal vessels became narrowed” to “The labyrinth formation was impaired and the areas of maternal vessels became narrowed” in the revised manuscript.  

  1. Explain the consequences of this finding: “In a total of 20983 genes, up-regulated genes were defined as those with both up-regulated in ZT0 and ZT12 (fold-change >1; 6147 genes) and selected by multiplying the absolute value of each WAD (>0.02; 597 genes), whereas down-regulated genes were those with both down-regulated (fold-change <1; 5169 genes) and selected (>0.02; 515 genes) as described previously [18]”.

We rewrote the next sentence “Gene ontology biological process term enrichment analyses of the identified up-and down-regulated genes detected by WAD were performed…..”.

Response to Referee II

Thank you for your thoughtful comments.

In response to your valuable advice, we extensively revised our manuscript as follows.

Title: "Bmal1" should be in italic.

We corrected the title in the revised manuscript.

Abstract:

1) In the beginning, it is better to add one or two brief sentences providing the background.

We added the sentence “Recently, it was demonstrated that the expression of BMAL1 was decreased in the endometrium of women suffering from recurrent spontaneous abortion” at the beginning of the abstract in the revised abstract.

2) Reproductive defects due to deletion of Bmal1 do not necessarily equal the function of the clock. That is, clock genes may have non-clock functions. This issue needs to be discussed in the manuscript. Of course, it is better to have experimental evidence to clarify.

We added this important issue prior to the conclusion paragraph in the Discussion section of the revised manuscript.  

Results

3) The occurrence of Fig.1 panels is not in order, Fig. 1D appears but Fig.1C is absent.

We added a description of Fig. 1C in the revised manuscript.

4) "cKO females increased incomplete miscarriage after day 6 of pregnancy", "pregnancy day 12", please phrase this sentence.

We corrected "pregnancy day 12" to " day 12 of pregnancy".

Novelty

5) I read the related paper (Proc Natl Acad Sci U S A. 2014 Sep 30;111(39):14295-300) and found that most of the findings in the present manuscript have been covered, including the phenotype description, the histological analysis, and supplementation of progesterone. Therefore, I strongly suggest that the authors provide more data on the molecular mechanisms. And this issue is also required to be clearly addressed in the discussion section.

As we described in the first paragraph, we clearly explained the apparent difference between our study and the previous study including reference #32 (Proc Natl Acad Sci U S A. 2014 Sep 30;111(39):14295-300). 

The study of reference #32 focused on ovarian function and demonstrated that deletion of Bmal1 in steroid hormone-producing cells in the ovary led to the dysfunction of the corpus luteum that produces progesterone. Ovarian progesterone is critical to achieving embryo implantation. Consequently, this mouse model showed the disorders in embryo implantation, which is clinically corresponding with implant failure. It is reasonable and natural that progesterone supplementation rescues embryo implantation. Therefore, they did not observe placental dysformation.

In contrast, our study focused on uterine function. Please note that in rodents, after embryo implantation, the function of corpora lutea of pregnancy is maintained by decidual prolactin in response to embryo implantation, and then by placental lactogen accompanied by placental formation. Using our mouse model, we firstly showed that the dysfunction of the uterine clock does not inhibit embryo implantation, but attenuates the maintenance of pregnancy and induces placental dysformation, which is clinically corresponding with miscarriage and placental dysfunction. To make it more comprehensive to understand by the readers, we cited the following papers that demonstrated clinical evidence of progesterone supplementation in the discussion section of the revised manuscript (Page 12). 

Ref. #42: Haas, D. M.; Hathaway, T. J.; Ramsey, P. S., Progestogen for preventing miscarriage in women with recurrent miscarriage of unclear etiology. Cochrane Database Syst Rev 2019, 2019, (11).

Ref. #43: Wu, H.; Zhang, S.; Lin, X.; He, J.; Wang, S.; Zhou, P., Pregnancy-related complications and perinatal outcomes following progesterone supplementation before 20 weeks of pregnancy in spontaneously achieved singleton pregnancies: a systematic review and meta-analysis. Reprod Biol Endocrinol 2021, 19, (1), 165.

6) "In accordance with it, this study showed that P4 supplementation", please remove the redundant space.

We deleted the redundant space in the revised manuscript.

7) Some important references are missing, which include:

Endocrinology 2016 Dec;157(12):4914-4929; Front Endocrinol. 2022 Mar 2;13:818272; J Biol Rhythms. 2008 Feb;23(1):26-36.

We added these important references at the beginning of the Discussion section in the revised manuscript. 

Round 2

Reviewer 1 Report

Accept in present form

Author Response

Response to Referee I

Accept in present form

Thank you for accepting our paper.

Response to Referee II

Thank you again for your thoughtful comments.

In response to your valuable advice, we revised our manuscript as follows.

  • Please rephase the sentences in discussion, for instance, like this way "Second, since clock genes may have non-clock functions, the deletion of uterine Bmal1 does not necessarily equal the dysfunction of the uterine clock system. In the future, it is necessary to validate this issue by experiments using other clock gene-deleted mice or by a constitutively expressed Bmail1 in the uterus".

We changed the sentences according to your advice.

  • Accordingly, change the last sentence in the abstract to be "These findings indicate that the uterine clock system may be critical for pregnancy maintenance after embryo implantation"

We changed the sentences according to your advice.

  • Some sentences and expressions need to be further polished. For instance, "disrupting a circadian rhythm", "time-restricted feeding directly regulates" "The circadian oscillator system is controlled by circadian clock genes", "produced conditional deletion of uterine Bmal1 (cKO) mice" etc. It is better to have the language improved by someone using English as native language.

We changed the sentences according to your advice.

Reviewer 2 Report

1) Please rephase the sentences in discussion, for instance, like this way "Second, since clock genes may have non-clock functions, the deletion of uterine Bmal1 does not necessarily equal the dysfunction of the uterine clock system. In the future, it is necessary to validate this issue by experiments using other clock gene-deleted mice or by a constitutively expressed Bmail1 in th uterus".

2) Accordingly, change the last sentence in the abstract to be "These findings indicatethat the uterine clock system may be critical for pregnancy maintenance after embryo implantation"

3) Some sentences and expressions need to be further polished. For instance, "disrupting a circadian rhythm", "time-restricted feedingdirectly regulates" "The circadian oscillator systemis  controlled  by circadian clock genes", "produced conditional deletion of uterineBmal1(cKO)mice" etc. It is better to have the language improved by someone using English as native language.

Author Response

(The authors gave the same response as above.)
